# Position: Age Estimation Models Do Not Process Biometric Data

**Nikita Marshalkin** [1]

## Abstract

When a neural network estimates someone's age from a photograph, does it process biometric data? The answer depends on whether identity-discriminative representations arise within the network during inference—a question that may seem trivial to ML researchers but triggers consent requirements under GDPR, statutory damages under BIPA, or high-risk AI classification under the EU AI Act. Yet no regulatory guidance addresses it. This position paper provides empirical evidence: 14 models evaluated across 3 face verification benchmarks show age estimators fall orders of magnitude short of identification thresholds. Age estimation models cannot identify individuals. We call on researchers to provide transparency about what systems store and can do, and on regulators to distinguish transient processing from template storage.

## 1. Introduction

When a neural network processes a photograph of a face to estimate someone's age, does it process "biometric data"? The answer determines whether operators must obtain explicit consent under the General Data Protection Regulation (GDPR) Article 9, face statutory damages of $1,000–$5,000 per violation under the Illinois Biometric Information Privacy Act (Illinois General Assembly, 2008), or comply with high-risk AI system requirements under the EU Artificial Intelligence Act (EU AI Act) (European Parliament and Council, 2024).

The question seems straightforward: age estimation models estimate age, not identity. However, GDPR Article 4(14) defines biometric data as data that "allows or confirms" unique identification (European Parliament and Council, 2016). During inference, neural networks compute intermediate representations that may encode facial geometry—though

[1]Sumsub GmbH, Berlin, Germany. Correspondence to: Nikita Marshalkin <nikita.marshalkin@sumsub.com>.

*Proceedings of the $43^{rd}$ International Conference on Machine Learning*, Seoul, South Korea. PMLR 306, 2026. Copyright 2026 by the author(s).

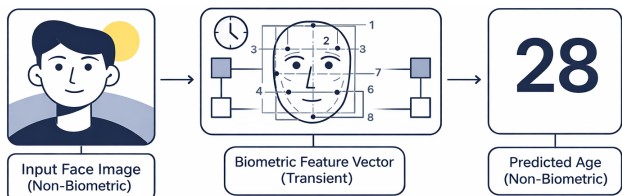

*Figure 1.* Age estimation pipeline. A face image enters, intermediate representations form transiently during inference, and only the predicted age exits. The regulatory question: do those transient intermediate representations constitute biometric data?

these representations exist only transiently and are never exposed; only the predicted age is output. If those representations contain identity-discriminative information, they might qualify as biometric data (Article 29 Working Party, 2012). This technical ambiguity creates regulatory uncertainty.

Regulators have not resolved this question. The UK ICO concluded that facial age estimation does not constitute special category data because it is not used "for the purpose of uniquely identifying individuals" (ICO, 2022). But the ICO's 2024 guidance acknowledges that biometric protections may apply "if it is possible to identify someone, even if not your intention" (ICO, 2024). The EDPB's 2019 guidelines carve out a "classification exception" for systems that detect physical characteristics without generating identity templates (EDPB, 2020), but the 2022 facial recognition guidelines take a broader stance (EDPB, 2023). Organizations deploying age estimation face unclear compliance requirements.

**Age estimation models cannot identify individuals.** They perform far below required thresholds on face verification benchmarks, falling two orders of magnitude short of regulatory requirements. Counterarguments exist: capability-based regulation serves legitimate purposes, and the precautionary principle counsels caution. But the empirical question has a clear answer. The contribution here is empirical evidence to inform regulatory deliberations, not to resolve the legal question.

This study probes the internal representations of 14 models for identity-discriminative capability across 3 face verification benchmarks, comparing against regulatory thresholds (NIST SP 800-63-4, EU Entry/Exit System, FIDO Alliance).

Layer-by-layer analysis shows that age estimation models have no meaningful verification capability at any layer. A secondary experiment trains attention-based probes on frozen features using identity-labeled data; even then, performance on harder benchmarks remains poor. The paper calls on the research community to provide transparency and on regulators to distinguish transient processing from template storage.

**Conflict of Interest Disclosure.** The author is employed by Sumsub GmbH, which develops the Commercial age estimator and Commercial sex estimator evaluated in this paper. The views expressed are solely those of the author and do not represent the position of any organization.

## 2. Background: The Regulatory Landscape

Three major regulatory frameworks define biometric data differently, and courts have drawn varying lines between classification and identification.

### 2.1. Definitions of Biometric Data

The GDPR establishes a two-tier framework. Article 4(14)[1] defines "biometric data" as "personal data resulting from specific technical processing relating to the physical, physiological or behavioural characteristics of a natural person, which **allow or confirm** the unique identification of that natural person" (European Parliament and Council, 2016). This is a capability-based definition: data qualifies as biometric if it is technically capable of enabling identification.

Article 9[2] then restricts processing of biometric data, but only when conducted "**for the purpose of** uniquely identifying a natural person" (European Parliament and Council, 2016). Biometric data processed for identification becomes "special category data," triggering stricter rules than ordinary personal data. This introduces a purpose-based limitation. The two tests differ: Article 4(14) asks whether data *can* identify someone; Article 9 asks whether data *is being used* to identify someone.

Recital 51 clarifies that photographs become biometric data "only when processed through a specific technical means allowing the unique identification or authentication of a natural person" (European Parliament and Council, 2016). The UK Information Commissioner's Office (ICO) elaborates that this transformation occurs through feature extraction and template generation. The ICO states that "any information produced by these specific technical processes, regardless of how long it exists for" constitutes biometric data

if capable of unique identification (ICO, 2024). The ICO further clarifies that "unique identification for the purposes of biometric data does not consider other sources of information you may hold or might be available. It is about whether someone can be directly identified from that information, with accuracy" (ICO, 2024). Closed-set context (for example, knowing a gallery contains only a few people) falls outside this test.

The Illinois Biometric Information Privacy Act (BIPA) takes a different approach (Illinois General Assembly, 2008). It defines "biometric identifier" as including "scan of hand or face geometry," explicitly excluding photographs but capturing the extraction of facial measurements. Unlike GDPR, BIPA focuses on whether facial geometry is extracted, not whether the purpose is identification. Courts have consistently held that extracting facial geometry from photographs triggers BIPA obligations regardless of intended use (Sosa v. Onfido, 2022).

The EU AI Act introduces yet another definition (European Parliament and Council, 2024). Article 3[3] defines biometric data (paragraph 34) as "personal data resulting from specific technical processing relating to the physical, physiological or behavioural characteristics of a natural person," omitting the identification requirement. The same article (paragraph 40) separately defines "biometric categorisation system" as "an AI system for the purpose of assigning natural persons to specific categories on the basis of their biometric data," explicitly listing age among such categories. Age estimation using facial analysis would fall under this definition if such processing produces biometric data—the very question this paper examines.

### 2.2. The Purpose Versus Capability Debate

The central interpretive question is whether technical capability or processing purpose should determine regulatory classification.

The **purpose-based interpretation** holds that Article 9's protections only apply when the controller's intent is unique identification. The ICO explicitly endorses this view: "This is specifically about the purpose you intend to use that information for" (ICO, 2024). In its regulatory sandbox with Yoti, a UK-based facial age estimation provider, the ICO concluded that facial age estimation "does not result in the processing of special category data" because "it's not being used for the purpose of uniquely identifying individuals" (ICO, 2022). While the biometric data question was explicitly raised during the sandbox, the ICO addressed only special category status under Article 9; whether age estimation produces biometric data under Article 4(14) was

---

[1]Article 4 of the GDPR defines key terms used throughout the regulation.

[2]Article 9 lists the "special categories" of sensitive data (health, religion, ethnicity, biometrics, and others) that require explicit consent or another specific legal basis to process.

[3]Article 3 of the EU AI Act provides definitions for the regulation, analogous to GDPR Article 4.

deemed "outside the scope" and remains unanswered.

The **capability-based interpretation** focuses on whether data is technically capable of enabling identification, regardless of stated purpose. The ICO's March 2024 guidance acknowledges this view: "if it is possible to identify someone, even if not your intention," the data may qualify as biometric (ICO, 2024). The US Federal Trade Commission takes this position: if it would be "reasonably possible to identify the person from whose information the data had been derived," it qualifies as biometric (FTC, 2023).

EDPB Guidelines 3/2019 on video devices address this directly (EDPB, 2020). Paragraphs 80-81 state that when a system "does not generate biometric templates in order to uniquely identify persons but instead just detects those physical characteristics in order to classify the person" (such as estimated age or gender), "the processing would not fall under Article 9." This classification exception has become the primary legal basis for age estimation systems in Europe.

EDPB Guidelines 05/2022 on facial recognition took a broader stance, stating that biometric data constitutes special category data for "both identification and authentication purposes" (EDPB, 2023). This creates tension with the earlier classification exception, and some national authorities now interpret the 2019 guidance more narrowly.

## 2.3. Enforcement Precedents

Enforcement actions have established clear precedents for facial recognition systems but left age estimation largely unaddressed.

The Clearview AI cases produced unanimous holdings across multiple jurisdictions that facial embeddings constitute biometric data. The Italian Garante imposed a EUR 20 million fine, finding that Clearview's facial recognition database processed biometric data without legal basis (Italian Garante, 2022). The French Commission Nationale de l'Informatique et des Libertés (CNIL) imposed a EUR 20 million fine, characterizing embeddings as "biometric templates... particularly sensitive, especially because they are linked to our physical identity" (EDPB, 2022).

These decisions addressed systems that stored facial embeddings for identification purposes. No known enforcement action has addressed transient intermediate representations from age estimation models.

In July 2025, CNIL declared age estimation cameras in French tobacco shops "neither necessary nor proportionate," prohibiting their use without resolving the biometric classification question (CNIL, 2025). The cameras analyzed all customers continuously; technical safeguards (local processing, no storage) were insufficient. This diverges from the UK ICO's Yoti sandbox, though the contexts differ: ICO addressed user-initiated online verification, CNIL passive retail surveillance.

In 2024–2025, regulators targeted adult website operators for inadequate age verification implementation (Ofcom, 2025). No formal enforcement actions were identified against age estimation providers.

## 2.4. The Regulatory Gap

Despite this regulatory activity, an open question remains: do transient intermediate representations from facial analysis models constitute biometric data?

The ICO confirms that biometric data "covers any information produced by these specific technical processes, regardless of how long it exists for" (ICO, 2024). But if intermediate representations exist only during forward propagation, are immediately discarded, and demonstrably cannot identify individuals, the classification becomes uncertain.

The ICO's Yoti sandbox addressed special category status (Article 9), finding that age estimation is "not used for the purpose of uniquely identifying" individuals (ICO, 2022). However, the report explicitly states that "a detailed assessment" of whether age estimation produces biometric data under Article 4(14) "falls outside the scope of this Sandbox project." The underlying capability question remains open. The following sections address that gap empirically.

## 3. Capability Versus Purpose

The purpose of age estimation systems is unambiguous: they estimate age. If purpose were the only criterion, age estimation clearly falls outside Article 9's scope because it is not conducted "for the purpose of uniquely identifying a natural person." The ICO's Yoti sandbox reached precisely this conclusion (ICO, 2022).

But the classification question has never been about purpose alone. The uncertainty lies in *capability*: whether the technical processing creates data that "allows or confirms" unique identification under Article 4(14), regardless of what the system is designed to do. Biometric data receives exceptional protection because of properties that exist independent of use: immutability (compromised biometrics cannot be reset), covert collection (capture without consent), and correlation with sensitive attributes. If capability exists, purpose becomes largely irrelevant to privacy harm.

The relevant test is usable identification capability: whether the system can single someone out, not whether identity-related information theoretically exists in intermediate representations. The ICO interprets "unique identification" as requiring "someone being singled out with accuracy . . . with a level of precision" (ICO, 2024). This precision is empirically testable—the contribution of Sections 5 and 6.

A credible counterargument holds that purpose-based interpretation better serves regulatory goals. The EDPB's classification exception in Guidelines 3/2019 reflects this view: systems that "just detect those physical characteristics in order to classify the person" should not face Article 9 restrictions (EDPB, 2020). Overly broad classification could impede legitimate age verification for child safety and burden developers with disproportionate compliance costs.

The purpose-based and capability-based approaches need not conflict. A system that cannot identify anyone poses no capability-based concern. The question is how to determine whether capability exists. Face verification benchmarks provide a test: if a model cannot determine whether two images depict the same person, it cannot identify individuals.

# 4. Standards and Regulatory Benchmarks

To assess identification capability empirically, reference points are needed. What performance levels do existing standards require for systems that are explicitly designed for biometric identification? These thresholds provide a concrete baseline against which to compare age estimation models.

## 4.1. Metrics for Biometric Verification

ISO/IEC 19795 establishes the standard terminology for biometric performance testing (ISO/IEC, 2021). For 1:1 verification (determining whether two samples come from the same person), two metrics are central:

**False Match Rate (FMR)**: The proportion of impostor comparison attempts incorrectly declared as matches. If the system says "same person" when comparing images of two different people, that is a false match. Also called False Accept Rate (FAR).

**False Non-Match Rate (FNMR)**: The proportion of genuine comparison attempts incorrectly declared as non-matches. If the system says "different people" when comparing two images of the same person, that is a false non-match. Also called False Reject Rate (FRR).

These metrics trade off against each other along a Detection Error Tradeoff curve. A system can achieve arbitrarily low FMR by rejecting everything, but FNMR will approach 100%. Conversely, accepting everything yields zero FNMR but 100% FMR; all impostor attacks will be approved as well. Useful systems operate at specific points on this curve, typically specified as "FNMR at FMR = X%." This notation means the false non-match rate when the system is tuned to achieve a particular false match rate.

*Table 1.* Biometric verification thresholds from major standards. FMR = security; FNMR = usability. Values are maximum acceptable error rates.

| Standard | FMR | FNMR | Context |
|---|---|---|---|
| NIST SP 800-63-4 | $\leq 0.01\%$ | $< 5\%$ | US identity |
| EU EES 2019/329 | $0.05\%$ | $< 1\%$ | EU borders |
| FIDO Alliance | $\leq 0.01\%$ | $< 5\%$ | Devices |

## 4.2. Regulatory and Industry Thresholds

Three major frameworks specify quantitative thresholds for biometric verification systems.

**NIST SP 800-63-4** (NIST, 2025) provides digital identity guidelines for US federal systems. Identity Assurance Level 2 (IAL2), required for most government identity proofing, mandates FMR $\leq 0.01\%$ (1:10,000) with FNMR $< 5\%$.[4] A demographic equity provision requires that no demographic group's error rate exceeds 25% worse than the overall population.

**EU Entry/Exit System (EES)** (European Commission, 2019) specifies mandatory thresholds for biometric verification at EU borders. For facial verification: FMR = 0.05% (1:2,000) with FNMR $< 1\%$. These are binding legal requirements for eu-LISA implementations.

**FIDO Alliance** (FIDO Alliance, 2025) certifies biometric authenticators for consumer devices. Certification requires FMR $\leq 0.01\%$ (1:10,000) at FNMR $< 5\%$, measured at 80% confidence upper bounds. Vendors may self-attest to thresholds as strict as 1:100,000.

Table 1 summarizes these requirements.

These thresholds define identification capability operationally. A system performing near random chance, orders of magnitude below these requirements, has no such capability.

# 5. Methodology

Face verification requires determining whether two face images depict the same person. We convert each image to an *embedding*, a vector that captures the face's distinguishing features, then compute the similarity between these embeddings. Higher similarity indicates that the images likely show the same individual. We evaluate performance using **FNMR@FMR** (False Non-Match Rate at a fixed False Match Rate), which measures how often genuine matches are incorrectly rejected when the false match rate is held constant.

Labeled Faces in the Wild (LFW) contains 3,000 impostor

---

[4]FMR $\leq 0.01\%$ is mandatory (SHALL); FNMR $< 5\%$ is recommended (SHOULD).

pairs. At FMR=0.01%, on average fewer than one impostor pair would exceed the threshold, making such measurements statistically unreliable. This study therefore reports primary results at FMR=1%. This operating point is $100\times$ more permissive than NIST/FIDO requirements, providing a conservative test of identification capability. Table 2 also reports FNMR@FMR=0.01% for comparison.

A model with no identity-discriminative information performs at chance; to achieve low FMR, such models must reject nearly all pairs, yielding FNMR approaching 100%. State-of-the-art face recognition achieves FNMR < 1% at FMR = 0.01% (Table 2).

### 5.1. Models Evaluated

The evaluation covers 14 models spanning age estimation, facial attributes, face recognition, and general-purpose vision, selected for diversity of architecture, training objective, and availability of pretrained weights. Table 2 summarizes the models and their verification performance.

**Age estimation models** are our primary subjects, trained to estimate age with no identification objective. **Facial attribute models** predict other characteristics (emotion, gender, liveness) for comparison. **Face recognition baseline** (InsightFace ResNet-100 with ArcFace) represents what true identification capability entails. **General-purpose vision models** (DINOv3, Perception Encoder, ImageNet ResNet-50) provide a baseline for incidental face representation from general visual learning.

### 5.2. Benchmarks

The evaluation uses three standard face verification benchmarks (Figure 2). **LFW** (Huang et al., 2008) provides 6,000 pairs of in-the-wild face images and serves as the standard baseline benchmark for face verification. Image quality varies: some pairs are well-lit studio photographs; others are low-resolution crops from news footage. It represents the easiest evaluation setting: if age estimation models cannot perform verification here, they certainly cannot do so on more challenging benchmarks. **AgeDB-30** (Moschoglou et al., 2017) tests pairs with 30-year age gaps, an adversarial setting probing whether age-related features aid or hurt verification. The same individual appears decades apart. **CFP-FP** (Sengupta et al., 2016) presents extreme frontal-to-profile matching, testing robustness to pose variation. Each pair consists of one frontal and one profile image of the same person.

### 5.3. Evaluation Protocol

For each model, we extract embeddings from intermediate layers, not just the final output, revealing where identity-discriminative information emerges. Global average pooling

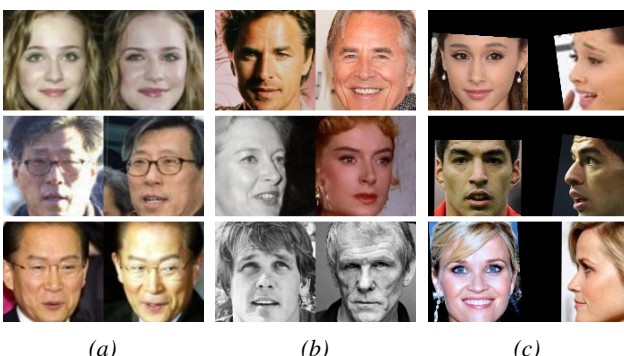

*(a)*      *(b)*      *(c)*

*Figure 2.* Sample pairs from face verification benchmarks. Each row shows a genuine pair. (a) LFW: in-the-wild images with varying quality. (b) AgeDB-30: same individual across a 30-year age gap. (c) CFP-FP: frontal and profile views.

is applied, embeddings are L2-normalized, and cosine similarity is computed between pairs. Average pooling is a minimal transformation that preserves identity information if present but cannot create it. For each layer and benchmark, we report FNMR@FMR=1% (primary, statistically reliable) and FNMR@FMR=0.01% (regulatory reference).

A secondary experiment tests whether identity information exists but is inaccessible via simple pooling. Models may allocate different information to different spatial locations or tokens (e.g., the CLS token in Vision Transformers), which average pooling would dilute. To test this, an attention pooler (Yu et al., 2022) is attached to the frozen backbone's final layer. The pooler uses a learned query to aggregate features via cross-attention. It then projects to an embedding space trained with ArcFace loss on the Glint360k dataset (An et al., 2021), the same dataset used to train the face recognition baseline. The probed model is no longer the age estimator: it is a new face recognition system that takes the age estimator's frozen features as input. This characterises what those features *contain*, not what the original age estimator computes during inference.

The evaluation uses established academic benchmarks. LFW, AgeDB-30, and CFP-FP contain images of public figures collected from news articles and Wikipedia. These benchmarks appear in hundreds of publications, enabling comparison with existing literature. The evaluation is read-only: embeddings exist transiently during the forward pass and are discarded after computing similarities. No templates are stored and no identification systems are built. The purpose is to demonstrate what these models cannot do, not to create identification capability.

## 6. Results

Age estimation models fail to distinguish individuals. Table 2 presents face verification performance across all mod-

*Table 2.* Face verification performance (FNMR%) at FMR=1% and FMR=0.01%. Lower is better. Age estimation models (bold) fall short by two orders of magnitude. Best layer shown for each model.

| Model | Arch. | LFW @1% | LFW @.01 | AgeDB-30 @1% | AgeDB-30 @.01 | CFP-FP @1% | CFP-FP @.01 |
|---|---|---|---|---|---|---|---|
| ArcFace (Deng et al., 2019) | ResNet-100 | 0.23 | 0.3 | 2.4 | 5.3 | 1.2 | 5.5 |
| Perception Encoder (Bolya et al., 2026) | ViT-B/16 | 20.8 | 56.7 | 97.6 | 99.8 | 79.7 | 97.6 |
| **Commercial age estimator** (Sumsub, 2026) | Proprietary | 26.8 | 63.7 | 97.2 | 99.9 | 80.4 | 97.8 |
| DINOv3 (Siméoni et al., 2026) | ViT-B/16 | 37.5 | 70.7 | 96.2 | 99.6 | 81.0 | 97.9 |
| **FairFace age+gender+race** (Kärkkäinen & Joo, 2021) | ResNet-34 | 57.4 | 85.8 | 98.4 | 99.9 | 82.3 | 98.7 |
| **Age+gender ViT** (Sahoo, 2025) | ViT-B/16 | 67.3 | 87.5 | 98.2 | 99.8 | 88.6 | 99.3 |
| **InsightFace age+gender** (InsightFace, 2018) | MobileNet 0.5 | 67.6 | 94.2 | 98.6 | 99.8 | 95.0 | 99.6 |
| **FaceLib age+gender** (Ayobi, 2020) | ShuffleNet V2 | 70.6 | 87.0 | 96.8 | 99.9 | 95.4 | 99.3 |
| Commercial sex estimator (Sumsub, 2026) | Proprietary | 75.5 | 86.6 | 95.6 | 99.1 | 96.2 | 99.6 |
| ImageNet classifier (He et al., 2016; Wightman) | ResNet-50 | 76.2 | 92.8 | 98.1 | 100 | 95.9 | 99.8 |
| Spoofing detector (Minivision Technology, 2020) | MiniFASNet | 85.4 | 96.7 | 97.6 | 99.9 | 97.8 | 99.9 |
| DeepFace emotion (Serengil, 2018) | VGG-like | 87.6 | 98.5 | 96.4 | 99.8 | 97.9 | 99.7 |
| **Age estimation PyTorch** (yu4u, 2019) | SE-ResNeXt50 | 94.6 | 99.7 | 98.2 | 100 | 97.8 | 99.8 |
| **SSR-Net** (Yang et al., 2018) | Compact CNN | 95.0 | 99.4 | 98.1 | 99.9 | 97.1 | 99.9 |

els and benchmarks, with LFW as the primary benchmark—the least adversarial setting, where failure precludes success on harder benchmarks.

Face recognition and age estimation models occupy distinct performance regimes. ArcFace achieves 0.23% FNMR at FMR = 1% on LFW. Age estimation models range from 27% (Commercial Age Estimator) to 95% (SSR-Net) FNMR at the same operating point. This gap spans two orders of magnitude. FMR=1% is already 100× more permissive than regulatory thresholds; at the regulatory threshold, age estimation models reach 64–99% FNMR.

AgeDB-30 provides a direct test of whether age-related features aid identity discrimination. Image pairs show the same person at ages up to 30 years apart. If age estimation models incidentally learned identity features through age variation, they should perform better here than generic models. Instead, they perform worse: all age estimation models saturate at 96–98% FNMR on AgeDB-30 at FMR=1%, compared to 27–95% on LFW. Features optimized for age estimation are orthogonal to identity-discriminative features.

Figure 3 shows FNMR across model layers on LFW for age estimation models. All models start near 95% FNMR in early layers. Some improve in later layers: the Commercial Age Estimator reaches 27% FNMR, FairFace 57%. Others show minimal improvement: SSR-Net and Age Estimation PyTorch remain above 90% FNMR throughout. Even the best age estimation layer remains far above the 5% FNMR requirement and two orders of magnitude worse than Arc-Face's 0.23%.

Cross-benchmark results reinforce this pattern. On AgeDB-30 and CFP-FP, all non-FR models saturate at 69–100%

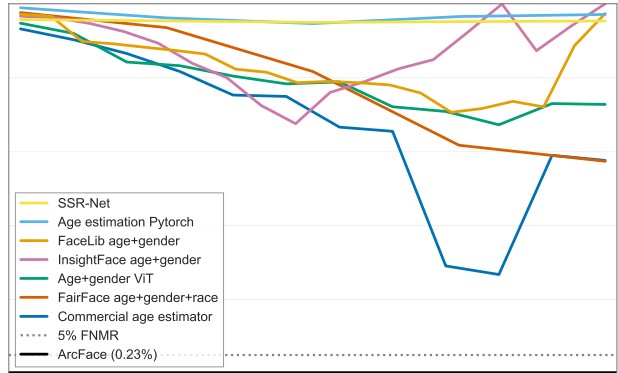

*Figure 3.* FNMR@FMR=1% across normalized model depth on LFW for age estimation models only (see Table 2 for all models). Lower is better. The dotted line marks 5% FNMR (NIST/FIDO threshold). ArcFace (solid black) achieves 0.23% FNMR. Age estimators start near 95% FNMR in early layers; even at their best, they remain far above thresholds.

FNMR regardless of layer. CFP-FP tests frontal-to-profile matching, a scenario less common in operational verification (border control captures frontal images). The consistency across benchmarks indicates these results are not artifacts of a particular dataset.

Table 3 shows results after attention probing with identity-labeled training. On LFW, all models improve, some substantially: the Commercial Age Estimator drops from 27% to 2% FNMR, Perception Encoder from 21% to 3%—yet

---

[5]Unlike all other models, this one performed better with average pooling and a linear layer than with attention pooling; we report the best result.

*Table 3.* FNMR@FMR=1% before and after attention probing. FMR=1% is 100× more permissive than NIST/FIDO (FMR=0.01%); at FMR=0.01% the strongest probed model still produces 17/91/68% FNMR on LFW/AgeDB-30/CFP-FP.

| Model | LFW | AgeDB | CFP-FP |
|---|---|---|---|
| ArcFace | 0.23 | 2.4 | 1.2 |
| Perception Encoder | 21 → 3 | 98 → 73 | 80 → 33 |
| Commercial age estimator | 27 → 2 | 97 → 67 | 80 → 28 |
| DINOv3 | 37 → 3 | 96 → 68 | 81 → 36 |
| FairFace | 57 → 20 | 98 → 97 | 82 → 74 |
| Age+gender ViT | 67 → 10 | 98 → 92 | 89 → 54 |
| InsightFace age+gender[5] | 68 → 60 | 99 → 99 | 95 → 96 |
| FaceLib age+gender | 71 → 35 | 97 → 98 | 95 → 94 |
| Commercial sex estimator | 76 → 29 | 96 → 97 | 96 → 87 |
| ImageNet classifier | 76 → 9 | 98 → 88 | 96 → 56 |
| Spoofing detector | 85 → 61 | 98 → 98 | 98 → 96 |
| DeepFace emotion | 88 → 88 | 96 → 98 | 98 → 98 |
| Age estimation PyTorch | 95 → 77 | 98 → 98 | 98 → 96 |
| SSR-Net | 95 → 81 | 98 → 98 | 97 → 96 |

still an order of magnitude worse than ArcFace's 0.23%. At the regulatory operating point (FMR=0.01%), the same probed Commercial Age Estimator reaches only 17/91/68% FNMR across the three benchmarks, well above the 5% threshold; LFW is reported at FMR=1% as the primary metric because its 3,000 impostor pairs make FMR=0.01% statistically less stable. On AgeDB-30 and CFP-FP, performance remains poor—28–98% FNMR even at FMR=1% after probing. Supervised training yields modest gains on less challenging benchmarks, but the representations lack the information needed for verification across age gaps or pose variations.

# 7. Alternative Views

Age estimation models cannot identify individuals. The debate is not settled.

## 7.1. The Capability and Precautionary Arguments

The ICO's March 2024 guidance states that biometric data protections may apply "if it is possible to identify someone, even if not your intention" (ICO, 2024). The FTC's 2023 Policy Statement similarly includes data where it would be "reasonably possible to identify the person" (FTC, 2023). Under this interpretation, even transient intermediate representations could constitute biometric data if they contain identity-discriminative information.

A related precautionary argument holds that regulatory uncertainty should favor protection. When harms are severe and irreversible, the prudent approach is to classify ambiguous cases as biometric.

The response is empirical. Capability-based regulation is legitimate. The argument is that the capability does not exist:

layer-by-layer analysis (Figure 3) shows no intermediate representation achieves meaningful verification performance. However, the precautionary principle should not be applied asymmetrically; the harms from over-regulating child safety systems are also real. Systems that cannot identify individuals can be excluded from biometric classification without abandoning precaution for systems that can.

CNIL's July 2025 ruling shows a third path: prohibition on proportionality grounds, bypassing biometric classification entirely (CNIL, 2025). However, the ruling targeted passive retail surveillance where customers could not opt out. User-initiated age verification, where individuals choose to verify for a specific transaction, involves a different proportionality analysis. CNIL's own 2022 guidance permits facial age estimation online when users consent and processing occurs locally.

## 7.2. The Function Creep Risk

A technical objection concerns potential misuse: an adversary with model weights could train a new head on frozen features to extract identity information. The EU AI Act addresses this through its "reasonably foreseeable misuse" framework (European Parliament and Council, 2024). A related framing holds that identity information may be latent in representations: if training a head on frozen features could extract identity signal, perhaps the representation "allows" identification under Article 4(14).

This argument proves too much. Original face images contain identity information (that is why face recognition works). Yet GDPR Recital 51 explicitly states that photographs are not biometric data until "processed through a specific technical means allowing the unique identification or authentication of a natural person" (European Parliament and Council, 2016). The distinction is not whether information theoretically exists, but whether specific technical processing has transformed it into a form that enables identification. If training a new model on frozen features made those features biometric data, the same logic would classify input images as biometric data, a position no regulator has adopted.

The attention probing experiment (Table 3) provides empirical support: even with the same training setup used for ArcFace, age estimation models fail on harder benchmarks. What matters is not whether an adversary could theoretically train a different system, but whether the age estimator's representations enable identification.

Function creep risk applies to any neural network; general-purpose vision models like DINOv3 contain face-discriminative features incidentally. Existing data protection laws address misuse through purpose limitation and security requirements, not by classifying all intermediate representa-

tions as biometric data.

## 7.3. The Clearview Precedent

Clearview AI enforcement actions across Europe produced unanimous holdings that embeddings constitute biometric data, accumulating over EUR 90 million in fines including EUR 20 million each from Italian and French authorities (Italian Garante, 2022; EDPB, 2022). If embeddings are biometric data, why would intermediate representations from age estimation models be different?

The distinction lies in what these systems do. Clearview stores embeddings in a searchable database of billions of images for explicit identification. Age estimation models do not store embeddings or build searchable databases; intermediate representations are discarded during inference. The Clearview precedent establishes that embeddings used for identification are biometric data. It does not establish that all intermediate representations in any facial analysis model are biometric data.

## 8. Call to Action

The classification of age estimation as biometric or non-biometric processing remains unresolved. EDPB Statement 1/2025 provides principles for age assurance but does not address whether facial analysis constitutes biometric data under Article 4(14) (EDPB, 2025). The EU Digital Omnibus Package distinguishes verification from identification but omits age estimation entirely (European Commission, 2025). ISO/IEC 27566-1:2025 establishes age assurance as a distinct domain without resolving the classification question (ISO/IEC, 2025). The ML research community can help clarify this landscape.

### 8.1. For the Research Community

When regulators ask whether age estimation processes biometric data, they are asking whether these systems can identify individuals. The answer depends on what the system stores and what its representations can do. During inference, an image enters, activations propagate through layers, intermediate representations form and dissolve, an output emerges. For age estimation, that output is an age prediction; the intermediate representations exist for milliseconds and are discarded. This differs from face recognition systems that extract embeddings for storage and later comparison.

The research community can communicate this distinction through documentation. Model cards could note whether embeddings are stored or discarded after inference. Technical reports could include verification benchmark results showing whether a model can distinguish individuals. Architecture descriptions could specify data flows: what representations are created, how long they persist, what they

can and cannot be used for. When a system processes a face transiently and outputs only an age estimate, that differs from a system that stores facial embeddings in a searchable database (Italian Garante, 2022; EDPB, 2022), and documentation should make this clear.

### 8.2. For Regulators

Current definitions leave key questions unanswered. GDPR Article 4(14)'s "allow or confirm the unique identification" does not specify whether transient intermediate representations qualify. Does a neural network activation that exists for milliseconds "allow" identification? Organizations building age verification for child safety face uncertainty about whether their systems trigger biometric data rules. When the legal boundary is unclear, organizations treat everything as worst-case. This impedes deployment of age verification where it could protect minors, and burdens systems with compliance requirements designed for identification technology they do not resemble.

One distinction would provide clarity: transient processing differs from template storage. Clearview AI stored facial embeddings in databases for later search; age estimation discards intermediate representations after inference. Guidance should acknowledge this difference. What matters is not what a representation could theoretically do, but what the system actually stores and does.

A practical criterion is the system boundary: what data leaves the pipeline. The ICO's Biometric Data Guidance already frames biometric recognition as a multi-stage pipeline, with feature extraction and template generation as distinct stages (ICO, 2024). The criterion applies at each interface: if a stage emits biometric features or templates, that data flow is auditable, even when the system's final output is not.

## 9. Conclusion

Do age estimation neural networks have identification capability? The evidence indicates they cannot. Across 14 models and 3 benchmarks, all evaluated age estimation architectures perform at or near chance on face verification. This falls two orders of magnitude short of regulatory thresholds. The claim is bounded by the architecture families tested: a model built on a frozen face-recognition backbone would retain identity by construction, but no model of that kind is in deployed age estimation use. If these models cannot distinguish one person from another, their intermediate representations do not meet GDPR Article 4(14)'s definition of biometric data at any operating point defined by existing biometric verification standards (see Table 1).

The argument is not that age estimation should be entirely exempt from privacy regulation. Facial images remain personal data. The question is whether specific biomet-

ric protections (Article 9 explicit consent, BIPA penalties, high-risk AI classification) should apply to systems that cannot identify anyone. Counterarguments exist, including capability-based interpretation, the precautionary principle, and function creep. These have been addressed above. Courts and regulators make legal determinations; the contribution here is evidence.

The approach provides an objective basis for classification decisions: extract embeddings from intermediate layers, evaluate on face verification benchmarks, and compare to regulatory thresholds. The classification has practical consequences. Overly broad classification impedes age verification for child safety. Overly narrow classification leaves identification systems inadequately regulated. The research community should provide transparency about what systems store and can do; regulators should develop guidance that distinguishes transient processing from template storage.

## Disclaimer

The author is an engineer, not a legal specialist. Nothing in this paper should be construed as legal advice. Readers facing regulatory questions should consult qualified legal counsel in their jurisdiction.

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
