# OpenReview forum: "Position: Age Estimation Models Do Not Process Biometric Data"
_ICML.cc/2026/Position_Paper_Track — ICML 2026 Position Paper Track regular_

### Official Review · Reviewer_USEw · 2026-03-04

**Significance:** 4
**Argument Clarity:** 4
**Rating:** 5
**Confidence:** 4

**Questions:**

1. In Table 1, you define the NIST and FIDO thresholds for “identification capability” as an FNMR of less than 5%. However, in Table 3, you report that after attention probing, your “Commercial age estimator” achieves a 2% FNMR on the LFW benchmark. Since 2% is more accurate than the 5% threshold you cited as the industry baseline for identification, how do you reconcile this result with your central thesis that these models lack the technical capability to identify individuals?

2. A primary recommendation of your paper is for regulators to distinguish “transient processing” from “template storage”. Yet, you cite the UK ICO’s 2024, which states that biometric data protections apply to any information produced by technical feature extraction “regardless of how long it exists for”. Given that a major regulator has addressed and dismissed the duration of data existence as a factor, what is your legal or logical rebuttal?

3. You conclude that if these models cannot distinguish individuals, their representations don’t meet the GDPR Article 4(14) definition “in any operational sense”. Could you clarify what you mean by “operational sense”? If a model falls short of the 1:10,000 FMR threshold by NIST but still provides a signal strong enough to distinguish individuals at a lower confidence level, does that not still technically allow identification under law?

**Alternative Views Section:**

Yes

**Compliance With Llm Reviewing Policy A Conservative:**

Affirmed.

**Discussion Potential:**

4

**Paper Summary:**

Contributions:
1. Study 14 different models across three face verification benchmarks (LFW, AgeDB-30, CFP-FP).
2. Performance results are compared against regulatory and industry thresholds for biometric verification (NIST SP 800-63-4, EES, FIDO Alliance).
3. Age estimation models fail to meet identification thresholds at any layer, falling two orders of magnitude short of regulatory requirements.
4. Attention-based probes were trained on frozen features using identity-labeled data; even then, performance on harder benchmarks remains poor on AgeDB-30 and CFP-FP.

Position advocated:
1. Age estimation models do not process biometric data because they lack the technical capability to uniquely identify individuals. The authors argue that these models perform near random chance on identity verification tasks and do not create the persistent template used in facial recognition systems.
2. Calls for a clear regulatory distinction between transient processing and template storage. It urges the research community to provide transparency through documentation, such as model cards as well as its data persistence.

**Position:**

Yes

**Position In Title:**

Yes

**Related Work:**

4

**Strengths And Weaknesses:**

Strengths:
1. Evaluation of 14 models (age, sex, emotion estimators) across 3 verification benchmarks.
2. Connects abstract legal definitions (GDPR Article 4(14)) to ML metrics like FMR and FNMR.
3. Show verification capability in practice (comparing performance against regulatory and industry thresholds from NIST, EES & FIDO Alliance).
4. Commitment to testing whether identity information is hidden rather than absent (supervised attention-based probing where we train a head on frozen features with identity labels).

Weaknesses:
1. LFW is standard for research, but might not truly reflect risks in billion-scale galleries where weak identity signals might enable some identification in a smaller, closed-set environment.
2. In Table 3, the “Commercial age estimator” achieves a 2% FNMR on LFW benchmark after probing. Because 2% is better than the 5% threshold set by NIST and FIDO, the model has the “technical capability” to identify individuals. Because GDPR Article 4(14) defines biometric data as anything that “allows or confirms” unique identification, it could be argued that the latent identity signal in the frozen features is enough to classify the processing as biometric.
3. The stated goal of the authors is to call “ regulators to distinguish transient processing from template storage”. So, the authors rely on a distinction that ICO has already declared irrelevant (“regardless of how long it exists for”). If the data is capable of identification (2% FNMR), the fact that it only exists for milliseconds doesn’t prevent it from being classified as biometric data.

**Support:**

4

---

> ### Author Rebuttal · Authors · 2026-03-29
>
> We thank Reviewer USEw. The operating-point mismatch (W2/Q1) caught a real presentation problem we will fix.
>
> **W1: Weak identity signals might suffice in closed-set environments.**
>
> The ICO's Biometric Data Guidance (2024) addresses this directly: "Unique identification for the purposes of biometric data does not consider other sources of information you may hold or might be available. It is about whether someone can be directly identified from that information, with accuracy." Closed-set context (knowing the gallery contains only a few people) is "other sources of information," excluded from the unique identification test. The question is whether the data itself singles someone out with accuracy. Consider a ten-person office: knowing someone is 63 may suffice to identify them. But then age itself is the identifier, not the model's internal representations.
>
> We will add this ICO clarification to Section 2.1.
>
> **W2/Q1: The 2% FNMR after probing appears to beat the 5% NIST threshold.**
>
> This points to a presentation problem we will fix. Table 3 reports probing results at FMR=1%, while Table 1 defines thresholds at FMR=0.01% - a 100x difference in error rate. At the correct operating point (FMR=0.01%), the probed Commercial Age Estimator achieves: LFW 17% FNMR, AgeDB-30 91%, CFP-FP 68%.
>
> We will add FMR=0.01% columns to Table 3 with a note: "The FMR=1% column is 100x more permissive than regulatory thresholds and should not be compared directly." We note that FMR=0.01% measurements on LFW are statistically less stable due to the 3,000 impostor pair limit, which is why we report FMR=1% as the primary metric. The direction is unambiguous.
>
> **W3/Q2: The ICO states protections apply "regardless of how long data exists for."**
>
> Duration is not our defense. The ICO's full statement includes a conditional: "It also covers any information produced by these specific technical processes, regardless of how long it exists for. If this information is capable of uniquely identifying someone, it is biometric data" (Biometric Data Guidance, 2024). Our experiments test that capability condition. The answer is no: age estimation representations cannot uniquely identify anyone at regulatory thresholds.
>
> Building on Section 8's discussion of transient processing versus template storage, we propose a concrete criterion: the system boundary. What matters is what the system outputs and stores. An age estimation system outputs a scalar prediction and retains no templates. Inspecting system outputs, API contracts, and data retention practices provides an auditable standard. We will add this to Section 8.2.
>
> **Q3: What does "operational sense" mean?**
>
> "Operational" means at any operating point where a system could function as an identifier. Assessing whether data "allows" identification requires a concrete quality threshold; we use existing biometric verification standards (Table 1) as our reference. At these operating points, age estimators achieve 64-99% FNMR.
>
> We will replace "in any operational sense" in Section 9 with: "at any operating point defined by existing biometric verification standards (see Table 1)."
>
> Concretely, we plan to: (1) add the ICO's unique identification clarification to Section 2.1, (2) add FMR=0.01% columns to Table 3, (3) add the system boundary criterion to Section 8.2, (4) replace "operational sense" with a precise definition in Section 9.

---

> > ### Author Rebuttal · Reviewer_USEw · 2026-04-01
> >
> > N/A

---

### Official Review · Reviewer_1Yew · 2026-03-13

**Significance:** 2
**Argument Clarity:** 3
**Rating:** 5
**Confidence:** 3

**Questions:**

NA

**Alternative Views Section:**

Yes

**Compliance With Llm Reviewing Policy A Conservative:**

Affirmed.

**Discussion Potential:**

2

**Final Justification:**

All my concerns have been addressed. I recommand acceptance.

**Paper Summary:**

This paper examines whether age estimation models process biometric data, a question the authors frame as consequential under GDPR, BIPA, and the EU AI Act.
The paper argues that the relevant technical issue is whether age-estimation systems generate identity-discriminative representations during inference.
To address that question, the authors evaluate 14 age-estimation models on 3 face-verification benchmarks and report that these models perform far below identification thresholds, including in layer-wise analyses and probing experiments.
Based on these findings, the paper advances the position that age-estimation models cannot identify individuals and therefore should not be treated as processing biometric data in the same way as identity-focused systems.

**Position:**

Yes

**Position In Title:**

Yes

**Related Work:**

2

**Strengths And Weaknesses:**

Strengths:

The theme studied by the article is how model capability should be interpreted in privacy regulation, and it solves the issue within that debate: whether age-estimation models should count as processing biometric data.
I like that the paper turns the question into an empirical evaluation.
In particular, the authors look at 14 age-estimation models, test them on 3 face-verification benchmarks, and also analyze intermediate layers rather than only final outputs.
That makes the paper feel much more grounded than a purely conceptual position paper.
I also found the additional probing experiment useful, since it directly addresses the obvious counterargument that identity information may still be latent in the features.

Weaknesses:

My main concern is that the paper sometimes states the conclusion a bit more broadly than the experiments really support.
The results do show that the evaluated models are far from practical face recognizers, but that is still narrower than saying age-estimation models in general do not process biometric data.
I would like to see a more careful discussion of scope: how much this depends on the chosen architectures, training data, and benchmarks.
Also, the probing results seem a little more mixed than the paper’s framing suggests.
If identity-related information can be recovered to some extent from frozen features, even if performance remains weak on harder benchmarks, then the boundary between "no identity capability" and "not useful for identification in practice" deserves a more precise discussion.

**Support:**

3

---

> ### Author Rebuttal · Authors · 2026-03-29
>
> We thank Reviewer 1Yew. The scope concern is fair and we address it below.
>
> **W1: Conclusions stated more broadly than experiments support.**
>
> The reviewer is right. The title is the position, not a universal proof. The 14 models cover the major architecture families in deployed age estimation: CNNs (ResNet, ShuffleNet, SE-ResNeXt), Vision Transformers, compact architectures (SSR-Net), and a commercial system. They were trained on different datasets with different losses and training procedures. This covers what is actually deployed. Not every conceivable age estimation model avoids processing biometric data - for example, one built on a frozen face recognition backbone would retain identity information by construction.
>
> The paper's second contribution is the methodology itself (Sections 5-6): extract embeddings, evaluate on face verification benchmarks, compare to regulatory thresholds. Section 9 already describes this. We will tighten the empirical claim in Section 9 from "age estimation architectures" to "all 14 evaluated age estimation architectures" so the scope is unambiguous, while keeping the methodology available for evaluating any future model.
>
> **W2: Probing results more mixed than framing suggests.**
>
> The reviewer is right that LFW improvement is substantial - the Commercial Age Estimator drops from 27% to 2% FNMR. But this required training a new system on frozen features: an attention head with ArcFace loss on 17M identity-labeled images. At that point, the system is no longer the age estimator - it is a new face recognition model using the age estimator's features as input. One could achieve better results training face recognition from scratch.
>
> The probing experiment characterizes what the features contain, not what the age estimator does during inference. GDPR Recital 51 draws the same line: photographs contain identity information yet are not biometric data until processed through means enabling identification. And the improvement on LFW does not hold at the regulatory operating point: FMR=0.01% vs. FMR=1% in Table 3. See our response to Reviewer USEw (W2/Q1) for the operating point distinction.
>
> We will revise the scope discussion in Section 9 and sharpen the probing framing.

---

> > ### Author Rebuttal · Reviewer_1Yew · 2026-04-02
> >
> > Given that the authors promise to make revision, the concerns are addressed.

---

### Official Review · Reviewer_2qc1 · 2026-03-13

**Significance:** 4
**Argument Clarity:** 3
**Rating:** 5
**Confidence:** 2

**Questions:**

1. Regarding the risk of “function creep,” the paper acknowledges that an adversary could fine-tune a new head on frozen features to extract identity information, but argues this does not make the original model a processor of biometric data. Could the authors quantify this risk more concretely? For instance, given access to a large dataset, how close could an attacker come to extracting near-identification-level representations from an age estimation model?

2. The paper qualitatively describes “transient processing” but does not propose technical metrics that regulators could adopt. Do the authors plan to develop quantifiable criteria (e.g., feature retention time, accessibility of intermediate representations, specific processing patterns) that could serve as a clear threshold for determining whether transient representations constitute biometric data?

**Alternative Views Section:**

Yes

**Compliance With Llm Reviewing Policy A Conservative:**

Affirmed.

**Discussion Potential:**

4

**Final Justification:**

I would like to maintain my initial rating.

**Paper Summary:**

This paper addresses a question of  policy relevance: when a neural network estimates a person’s age from a photograph, does it constitute processing of biometric data? By using technical experimentation and regulatory analysis, the authors systematically evaluate 14 models across three standard face verification benchmarks (LFW, AgeDB-30, CFP-FP). The results show that age estimation models perform orders of magnitude worse than regulatory thresholds (e.g., NIST, FIDO, EU EES) on identity verification tasks. The paper further examines definitions of biometric data under frameworks such as GDPR, BIPA, and the EU AI Act, highlighting a regulatory gap concerning whether “transient intermediate representations” qualify as biometric data. Based on empirical evidence, the paper argues that age estimation models should not be classified as processing biometric data, and calls on researchers to provide model transparency and on regulators to distinguish between transient processing and template storage. The “Alternative Views” section addresses and rebuts opposing stances, including capability-based interpretations, the precautionary principle, and function creep concerns.

**Position:**

Yes

**Position In Title:**

Yes

**Related Work:**

4

**Strengths And Weaknesses:**

**Strengths**:
1. The topic is interesting and important. The paper tackles a timely and impactful issue—whether age estimation constitutes biometric data processing—which remains unresolved in current regulations. It fills a critical gap between technology and law, offering valuable insights for industry compliance and regulatory policymaking.
2. Experiments are supportive. The evaluation covers 14 diverse models (age estimators, attribute classifiers, general-purpose vision models) on multiple benchmarks, with layer-wise analysis and attention probing experiments. This rigorous methodology enhances the credibility of the findings.

**Weaknesses**:
1. The paper sometimes equates “identification capability” with performance on verification tasks, and at other times with the presence of identity information in representations. It does not clearly distinguish between “information existence” and “information usability,” which is central to the legal debate.
2. Logical gap between empirical results and policy claim: While the models perform poorly on verification, the paper does not sufficiently justify why poor verification performance equates to “not processing biometric data.” The legal definition may focus on “possibility of identification” rather than “successful identification.” A clearer delineation between capability and intent is needed.

**Support:**

3

---

> ### Author Rebuttal · Authors · 2026-03-29
>
> We thank Reviewer 2qc1. The existence/usability distinction (W1) is the right question and sharpened our argument.
>
> **W1: Not clearly distinguishing "information existence" from "information usability."**
>
> The paper's position rests on usability, not existence. Section 2.1 makes this argument through Recital 51: photographs contain identity information, yet GDPR states they are not biometric data until "processed through a specific technical means allowing... unique identification." The ICO elaborates: "It's only when something else happens to that photo - a discrete processing operation... that results in something that allows or confirms someone's unique identification - that the result becomes biometric data" (Biometric Data Guidance, 2024).
>
> We will add to Section 3: "The relevant test is usable identification capability: whether the system can single someone out, not whether identity-related information theoretically exists in intermediate representations."
>
> **W2: Logical gap between empirical results and policy claim.**
>
> The ICO interprets "unique identification" through UK caselaw as "someone being singled out with accuracy (ie where they are distinguished from others with a level of precision)" (Biometric Data Guidance, 2024). Precision is measurable. Our experiments measure it. We acknowledge mapping Article 4(14) to certification thresholds is an interpretive contribution. But without any capability threshold, Article 4(14) would classify virtually any facial processing as biometric - including computing average pixel brightness.
>
> We will add to Section 3: "The ICO interprets "unique identification" as requiring "someone being singled out with accuracy... with a level of precision" (Biometric Data Guidance, 2024). This precision is empirically testable - the contribution of Sections 5-6."
>
> **Q1: Quantifying function creep risk.**
>
> The probing experiment directly quantifies this. Even with best-case conditions (frozen features, 17M identity-labeled images, ArcFace loss, a learned attention head), the probed model fails all benchmarks at the regulatory operating point, FMR=0.01%. See our response to Reviewer USEw (W2/Q1) for full numbers.
>
> One could go further - attach a larger extraction network or tap earlier layers with more raw information. But an adversary with those resources can train face recognition from scratch on raw images. The age estimator is an unnecessary middleman. The risk lies in access to identity-labeled data and compute, not in the age estimator.
>
> Section 7.2 already makes this argument: if exploiting frozen features made them biometric data, the same logic would classify raw photographs - a position no regulator has adopted.
>
> **Q2: Technical metrics for regulators.**
>
> Section 8 discusses transient processing versus template storage. This prompts a concrete criterion: the system boundary. An age estimation system outputs a scalar prediction and retains no templates. A face recognition system outputs an embedding for storage and comparison. If only non-biometric outputs leave the pipeline, the system is auditable regardless of internal architecture.
>
> We will add to Section 8.2: "A practical regulatory criterion is the system boundary: what data leaves the processing pipeline? Inspecting system outputs, API contracts, and data retention practices provides an auditable standard." The ICO's Biometric Data Guidance already frames biometric recognition as a multi-stage pipeline, listing feature extraction and template generation as distinct "main stages involved in biometric recognition systems." For composite pipelines, the criterion applies at each interface: if any stage produces biometric features or templates, that data flow is auditable - even if the system's final output is not biometric.
>
> We will revise the paper to: (1) clarify the existence/usability distinction and add the ICO precision interpretation in Section 3, (2) add the system boundary criterion to Section 8.2.

---

> > ### Author Rebuttal · Reviewer_2qc1 · 2026-04-02
> >
> > N/A

---

### Decision · Program_Chairs · 2026-04-30

**Decision:**

Accept (regular)

**Comment:**

All reviews were uniformly supportive of the paper.
The alternative views are widespread and may directly impact on how machine learning can be legally used today.
The paper has strong experimental support which clearly makes the case for the fact that the features used by age verification models cannot be used for identification.

From a legal perspective, I had two minor concerns that I hope the authors will clarify the final version of the paper:

1. The discussion of the AI act is a little confusing. If you want to argue the outputs of age estimation are not biometric data you would have to show that this age is not "personal data ... relating to the ... physical ... characteristics of a natural person". I strongly encourage the authors to clarify this. The results of the paper would be interesting in their own right, even if they only cover the GDPR and not the AI act.
2. The discussion of transitory data is also unclear. If the data has been destroyed, why is it possible to identify someone using it?